# An Analysis of the Goat Value Chain from Lao PDR to Vietnam and a Socio-Economic Sustainable Development Perspective

Thi Nga Bui [1,*] , Huu Van Nguyen [2], Xuan Ba Nguyen [2], Van Nam Le [3], Thi Mui Nguyen [2], Cuc Thi Kim Ngo [4,5], Quyen Thi Le Ngo [4], Nam Hoang [6], Luis Emilio Morales [6] , Viet Don Nguyen [5,7] , Luisa Olmo [7], Stephen Walken-Brown [7] and Thi Thu Huong Le [8]

1    Faculty of Tourism and Foreign Languages, Vietnam National University of Agriculture,
     Hanoi 100000, Vietnam
2    Faculty of Animal Sciences and Veterinary Medicine, University of Agriculture and Forestry,
     Hue University, Hue 49000, Vietnam; nguyenhuuvan@huaf.edu.vn (H.V.N.);
     nguyenxuanba@huaf.edu.vn (X.B.N.); nguyenthimui@huaf.edu.vn (T.M.N.)
3    Faculty of Rural Development, University of Agriculture and Forestry, Hue University,
     Hue 49000, Vietnam; levannam@huaf.edu.vn
4    National Institute of Animal Science, Hanoi 100000, Vietnam; cucngokim@yahoo.com (C.T.K.N.);
     ngothilequyen.nias@gmail.com (Q.T.L.N.)
5    Faculty of Animal Science, Vietnam National University of Agriculture, Hanoi 100000, Vietnam;
     nvietdon@gmail.com
6    UNE Business School, University of New England, Armidale, NSW 2351, Australia;
     nam.hoang@une.edu.au (N.H.); lmorales@une.edu.au (L.E.M.)
7    School of Environmental and Rural Science, University of New England, Armidale, NSW 2351, Australia;
     luisa.olmo@une.edu.au (L.O.); swalkden@une.edu.au (S.W.-B.)
8    Faculty of Accounting and Business Management, Vietnam National University of Agriculture,
     Hanoi 100000, Vietnam; thuhuonglt.qtkd@gmail.com
*    Correspondence: hieu0306@gmail.com; Tel.: +84-(0)9-1883-9181

**Abstract:** Goats are produced in Laos on a small scale and most of them are exported to Vietnam, where they receive a price approximately 30% higher than Vietnamese crossbred goats. In 2021, Laos exported 2.2 million dollars in sheep and goats, mostly to Vietnam (2.19 million dollars). Vietnam was also the fastest-growing export market for sheep and goats of Laos between 2020 and 2021, comprising 66.7% of the total exports. This study aims to analyze the goat value chain from Laos to Vietnam and, based on its characteristics, to suggest policy interventions for the socio-economic sustainable development of the chain. This research analyzes a sample of 400 survey interviews of goat chain actors collected during the second and third quarters of 2022, with the support of CommCare software version 2.53.1. The results show that the Lao goat chain has four main functions implemented by four prominent actors: goat-rearing farmers, traders, abattoir owners, and restaurant owners. However, the role of input suppliers is unclear in this chain. Support to the stakeholders has been provided by local authorities and the government, projects and NGOs, technical supporters, and license supporters. There was not any clear evidence of the goat processing industry. Vertical and horizontal linkages between stakeholders exist, but agreements are primarily oral. There is no declaration of leading actors, and a traceability system has yet to be implemented. The increase in Lao goat exports to the Vietnamese market is driven by the high demand from Vietnamese consumers. The commercialization of goats' yields provides a positive net income in Lao, where all actors benefit, especially traders and restaurant owners. The chain also creates job opportunities and income that improve living standards, especially for disadvantaged groups, such as women, middle-aged people, people with low literacy, and those living in rural areas. For the sustainable development of the chain, this research recommends that the Lao and Vietnamese governments work together to develop more favorable conditions for goat trading, to improve the traceability across the goat chain, to promote goat husbandry and feeding practices, and to foster goat farmer collaboration by sharing goat-rearing experiences.

**Keywords:** goat rearing; goat meat; small scale production; traders; marketing; abattoirs; restaurants

## 1. Introduction

Lao People's Democratic Republic (Lao PDR) is a landlocked country at the heart of the Indochinese Peninsula in Southeast Asia. It shares 505 km of border with China to the north, 435 km of border with Cambodia to the south, 2069 km of border with Vietnam to the east, 1835 km of border with Thailand to the west, and 236 km border with Myanmar to the northwest [1]. The country can be divided into lowland and upland/sloping land zones. Lowland areas—called the Mekong Corridor—are mainly along the Mekong River. Agriculture in the Mekong Corridor areas is becoming more and more market-oriented, with market forces driving the process of agricultural intensification and diversification. Upland villages are more remote and have poorer road and market access, and villages rely predominantly on subsistence farming [2].

Agriculture plays an important role in the economy as it creates employment for about one-fourth of the labor force, generates a source of income, and is an essential component of international trade in Lao. In addition, most of the agricultural and livestock production comes from smallholders. Livestock are an integral part of smallholder farming systems with over 95% of livestock being produced by smallholders [2]. Among livestock, small ruminants including goats are important for the livelihood of smallholders as a means of meat consumption and hide and skin production [3], providing a way to offset the risks of crop failure and diversification [4]. It is also a means of accumulating assets [5], earning cash income [6], and providing draft power and manure for crops [7]. Goat rearing also has the advantages of lower feeding costs, herds that are easier to manage, quicker capital rotation, and more suitable production that can be slaughtered at the household level [8,9]. Given the conditions in Laos, goats can develop with minimal management [10,11]. With 18,761 thousand hectares of forest, accounting for 81.3% of the land area [2], Lao has good and favorable conditions for ruminant development, including goat rearing.

Although goat production in Vietnam has developed fast at around 13.0% annually, with a total population of 2,675,188 by June 2022 [12], Vietnam has imported goat meat valued at 426,000 dollars in 2017 [13], mainly from Lao PDR due to the high demand of goat meat for consumption [14]. Research under SRA LPS/2016/027 [15] found that although goat production in Laos is small-scale, it is mainly produced for sale (94%), and up to 90% of the goats produced in the surveyed regions of Lao were exported to Vietnam. The goats in Vietnam received a price that was 30% higher on average than the price paid for the Vietnamese crossbred goats. In 2021, Laos exported 2.2 million dollars in sheep and goats, and the main destination was Vietnam with 2.19 million dollars, which accounted for 99.6% of the total value. Vietnam was also the fastest-growing export market for sheep and goats of Laos between 2020 and 2021 at 66.7% (877,000 dollars) [16].

Lao goats have a high potential for further development through a demand-driven chain originating from Vietnamese customers. The study by NAFRI, NAFES, and NUOL [17], Hoang et al. [18], and Gray et al. [15] reported that there is a potential for goat production and marketing in Laos. However, significant work has not been identified that describes the goat chain in Laos in general and the goat value chain from Laos to Vietnam in particular. How does the goat value chain from Laos to Vietnam operate? Can goat-rearing smallholders in Laos and other actors benefit from the chain? To understand this case study, this research has been implemented with the aims of analyzing the goat chain and finding out the function, operating activities, and value-added distribution along the goat chain from Laos to Vietnam and suggesting policy recommendations to improve the benefits and sustainability of the whole chain.

We proceeded by first introducing and providing a literature review of the goat value chain analysis. Then, we described the study sites, sampling method, data collection, and processing. After that, we analyzed the results and discussion. We finalized with a conclusion.

*Literature Review*

The value chain was first introduced and described in 1985 by Michael Porter [19]. After that, many scientists have studied and used the value chain as a tool for research [20–22]. Most of the scientists defined a value chain as a full range of activities from the different phases

of production to bring a product or service to consumers [21–27]. Other authors focused on specific aspects of the value chain, such as Gereffi [20] who concentrated on the significance of the quasi-hierarchical type of governance of the value chain, while Humphrey and Schumitz [28] emphasized the linkages among the various actors along the chain. According to John [29], value chains contribute to gathering all the factors of production and economic activities to formalize and supply a product to a destination. Lin et al. [30] found that in the case of uncertainty, uncertainty analysis, and optimization modeling were widely applied, particularly to support decision-making.

In 1994, a global approach was introduced to analyze the global integration of firms and countries and the determinants of global income distribution. At the same time, they presented the global commodity chain, which concentrated on the coordination of global dispersion linked with production systems. Based on that, the global value chain concept was developed and attention was paid to the governance structure of value chains [26].

Many scientists and organizations did research and agreed with the five main stages of a value chain: input suppliers; production; assembly; processing; and logistics [21,22,29,31–33], proving that the value chain is a good way for small businesses in developing countries to obtain upgrading information and enter the global market. The GTZ-SME [34] developed a value chain approach to become a tool for the development of small and medium-sized enterprises. In addition, Martin [35] indicated that a value chain approach could be a tool to improve corporations' performance. Furthermore, Ivarsson and Alvstam [36] and Jacques [37] proved that value chains could provide support by introducing new forms of production, technologies, logistics, labor processes, and organizational relations and networks. Moreover, Martin [35] showed that value chain-based approaches can be used as tools for governments and donors to support links, facilitate upgrading for greater returns, and enhance foreign direct investment (FDI) programs. According to Mitchell, Coles, and Keane [38], many international organizations in the development sector could use the value chain as a tool to identify poverty reduction strategies. However, there have been some reports indicating the value chain approach has some shortcomings [24], as it can create confusion due to overlapping terminology without clear definitions of the concepts and an insufficiently defined theoretical framework.

Kaplinsky and Morris [21] and Martin [35] indicated that value chain analysis focused on describing the dynamic linkages between productive activities and chain actors and examined the added value along the chain. The process of analyzing the value chain included the analysis of the economic costs; the calculation of the value added at each chain level; finding out the leading actors, bottlenecks, and the institutional framework of the value chain; and a potential for market growth [39]. In addition, Kaplinsky and Morris [21] found that value chain analysis is valuable for new producers, especially poor producers and poor countries to achieve sustainable income growth.

The five major stages of analyzing value chains include sector choice, market analysis, value chain mapping, a performance measure and benchmark, and analysis of the performance gap [22,31,32,40,41]. Among different approaches of value chain analysis, some major ones that are well-known are summarized in Table 1.

**Table 1.** Value chain analysis processes of some major theories.

| M4P | WUR | ACDI/VOCA | Raphael Kaplinsky and Mike Morris | GTZ |
|---|---|---|---|---|
| Tool 1—Prioritizing value chains for analysis | | End market opportunities | The point of entry for value chain analysis | Selecting a value chain for promotion |
| Tool 2—Mapping the value chain | Step 1—Map the value chain and its actors | Business and enabling environment | Mapping value chains | Value chain mapping |
| Tool 3—Governance, coordination, regulation, and control | Step 2—Identify key institutional factors influencing value chain | Vertical linkages | Product segments and critical success factors in final markets | Quantifying and analyzing value chains in detail |

**Table 1.** *Cont.*

| M4P | WUR | ACDI/VOCA | Raphael Kaplinsky and Mike Morris | GTZ |
|---|---|---|---|---|
| Tool 4—Relationships, linkages, and trust | Step 3—Synthesis of drivers, trends, and issues | Horizontal linkages | How producers access final markets | Economic analysis of value chains |
| Tool 5—Demand-driven upgrading, knowledge, skill, technology, and support service | Step 4—Explore future scenarios and visions | Supporting markets | Benchmarking production efficiency | Agreeing on a vision and strategy for value chain upgrading |
| Tool 6—Analyzing costs and margins | Step 5a—Identify key opportunities, barriers, and underlying causes | Upgrading | Governance of value chains | Analyzing opportunities and constraints |
| Tool 7—Analyzing income distribution | Step 5b—Identify options to overcome barriers and build on opportunities | Inter-firm cooperation | Upgrading in value chains | Setting operational upgrading objectives |
| Tool 8—Analyzing employment distribution | Step 5c—Cluster options and specify institutional implications and actions | Transfer of information and learning between firms | Distributional issues | Identifying actors to implement the upgrading strategy |
| | Step 6—Identify strategies for supporting/driving change | Power exercised by firms in their relationships with each other | | Anticipating the impact of chain upgrading |
| | | | | Facilitating the chain development process |
| | | | | Strengthening private business linkages |

Source: Khai [42].

Mohamadou [43] indicated that goat value chains use inputs and services to create added value along the chain from live goat production to transport, process and marketing, and the consumption of goat and goat meat products. Chowdhury et al. [44] analyzed a goat value chain in Bangladesh and pointed out that no value-added products/processed meat appeared in the market and the profit was much lower than that of the middleman, and he suggested improving the husbandry practices and reducing the production cost.

In another work, Islam et al. [45] showed that women and children are mostly reared in smallholder farms with 2–5 heads. In rural areas, nearly three-fourths of goats were reared with mostly natural grass and tree leaves. Furthermore, Siddiky [46] pointed out that the value chain of goats in Bangladesh was traditional and had poorly organized marketing channels from production to consumption, with many middle actors such as collectors, traders, and butchers. Chowdhury et al. [44] evaluated the socio-economic aspects of goat farmers and showed that goat farmers were middle-aged with a low level of education and high years of experience and women were involved in decision-making of goat production and marketing. Research by Sarker et al. [47] reported the domestic distribution channel of goats in Bangladesh with a lot of collectors, traders, and butchers. The income received by farmers varied and depended on the management procedure and selling time; hence, the economic benefit received by farmers was positive but quite low.

Research on goat value chains in Bangladesh by Barua et al. [48] found that farmers had little knowledge of goat farming, and the actors were not working well while the stakeholders did not supply enough support for a standard goat chain. Dube et al. [49] analyzed the goat value chain in Zimbabwe and showed that the goat chain included six major functions from input supply, production, trading, slaughter/processing, retailing, and consumption with many stakeholders to support the chain. Being free of disease and the animal's condition, age, weight, and the color of its meat were the most important attributes of value chain actors. Producers faced many challenges in the development process. According to Hussen et al. [50], in the goat value chain in Ethiopia, there were five major functions in the chain from input supply, production, marketing, processing, and

consumption. There were five domestic marketing channels with three live goat export and four goat meat export channels. The vertical linkages were mainly based on verbal contracts while the horizontal linkages were limited among export abattoirs and small traders. Export abattoirs seemed to have the most power along the chain.

Although there are a lot of studies about value chains in general and goat value chains in particular, none of them focuses on goat value chains from Laos to Vietnam. Therefore, there is a need to determine how goat value chains from Laos to Vietnam work. In addition, this research contributes to the literature by analyzing if goat chain actors get economic and social benefits from the chain.

## 2. Materials and Methods

### 2.1. Study Sites

Three provinces of Laos (Khammounane, Luang Prabang, Savannakhet) that produce the highest number of goats [15] and share a border and have regular goat exchange with seven provinces of Vietnam (Dien Bien, Son La, Nghe An, Ha Tinh, Quang Binh, Quang Tri, Thua Thien Hue) were chosen to conduct the survey. In addition, Vientiane, the capital and the largest city of Laos, and three provinces with a high-consumption demand for goats in Vietnam, including Hanoi, Hoa Binh, and Da Nang, were also chosen to explore the destination for the Lao goat chain (Figure 1).

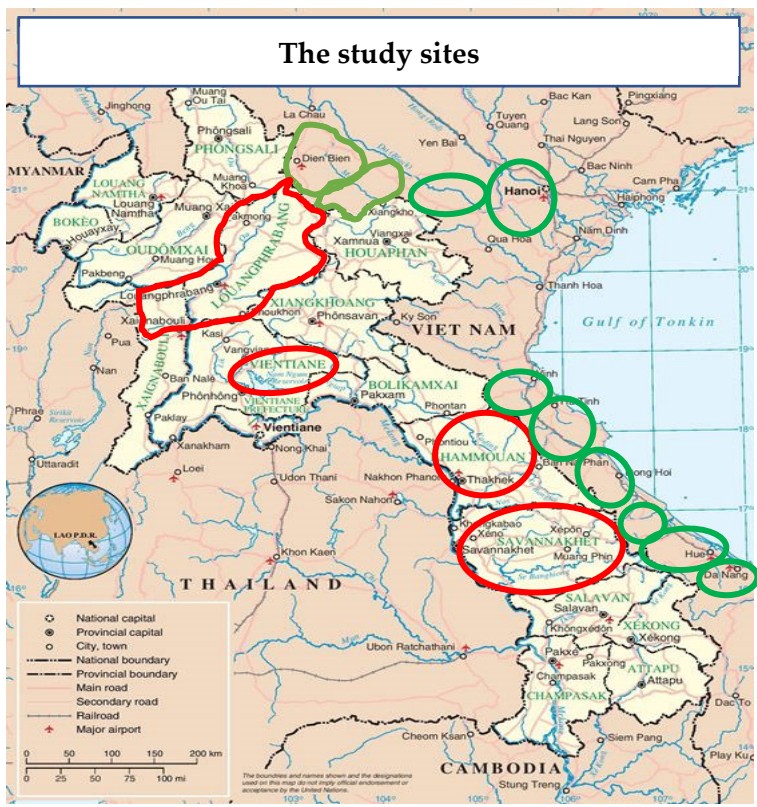

**Figure 1.** The study sites. Source: Laos Map [51]. Note: Oval shapes highlighted in red are study sites in Laos; Oval shapes highlighted in green are study sites in Vietnam.

### 2.2. Sampling Method

For farmers, the sample size (SS) was selected based on the non-probability sampling method [52] with specific sample numbers

$$SS = \frac{Z^2 \times (p(1-p))}{e^2} = \frac{1.96^2(0.05(1-0.05))}{0.03^2} = 202.75$$

where Z is the standard distribution statistical value. With 95% confidence, Z = 1.96.

p: the probability of selection. With the limitation of study time, we chose p = 5%.

*e*: level of error, e = 3%.

The target survey group is goat-rearing farmers. In total, 221 farmers were interviewed. After cleaning the data, they were used for processing and analysis.

For other actors, all abattoirs, all possible traders, and goat restaurants or restaurants that have goat dishes in the surveyed provinces were interviewed.

### 2.3. Data Collection and Processing

To collect the primary data, four structured questionnaire sets were designed to interview the four prominent chain actors, including goat-rearing farmers, traders, abattoirs, and restaurant owners. A pre-test of the questionnaire sets was conducted before finalizing the formal version. The final questionnaire sets were used to collect qualitative and quantitative data for the analysis [53]. The interview was conducted with the chain actors in the second and third quarters of 2022.

Additionally, informal conversational interviews were exploited to explore a broad field related to goat farming and marketing [54] and gain insight into problems in the chain. In addition, in-depth interviews were also used to provide more qualitative and quantitative information on the different participants and to understand their linkages. Moreover, observation [54] was used to crosscheck the information obtained from other sources, collecting related information about infrastructure, hygienic conditions, and the attitudes of those interviewed.

Information was collected from 400 respondents, 265 from Laos and 135 from Vietnam, of which there were 221 farmers in Laos, including Khammouane (59), Luang Prabang (65), Savanakhet (83), Viettiane (14); 36 traders: 7 in Laos and 29 in Vietnam; 36 slaughterhouse owners: 15 in Laos and 21 in Vietnam; and 107 restaurant owners: 22 in Laos and 85 in Vietnam (Table 2).

After cleaning, the data were processed using Stata 14 software and presented in tables and figures.

**Table 2.** Sample size of respondents.

| Country | Province | Farmer | Trader | Abattoirs | Restaurant Owner | Total |
|---------|----------|--------|--------|-----------|------------------|-------|
| Laos | Khammouane | 59 | 1 | 2 | 4 | 66 |
| | Luang Prabang | 65 | 3 | 2 | 5 | 75 |
| | Savannakhet | 83 | 2 | 6 | 8 | 99 |
| | Vientiane | 14 | 1 | 5 | 5 | 25 |
| Laos total | | 221 | 7 | 15 | 22 | 265 |
| Vietnam | Da Nang | | 1 | | 4 | 5 |
| | Dien Bien | | 5 | 5 | 6 | 16 |
| | Ha Tinh | | 3 | | 10 | 13 |
| | Hanoi | | | 7 | 7 | 14 |
| | Nghe An | | 2 | | 9 | 11 |
| | Hoa Binh | | 4 | 4 | 10 | 18 |
| | Quang Binh | | 1 | | 10 | 11 |
| | Quang Tri | | 5 | | 10 | 15 |
| | Son La | | 5 | 2 | 9 | 16 |
| | Thua Thien-Hue | | 3 | 3 | 8 | 14 |
| Vietnam total | | | 29 | 21 | 85 | 135 |
| Total | | 221 | 36 | 36 | 107 | 400 |

Source: Survey data, 2022.

*2.4. Data Analysis*

The methodology of commodity chain analysis [55] was used for financial analysis based on the value added (VA). VA was calculated by the equation:

$$VA = TO - IC$$

where TO represented turnover, and IC was the intermediated cost, as presented in the following Chart 1:

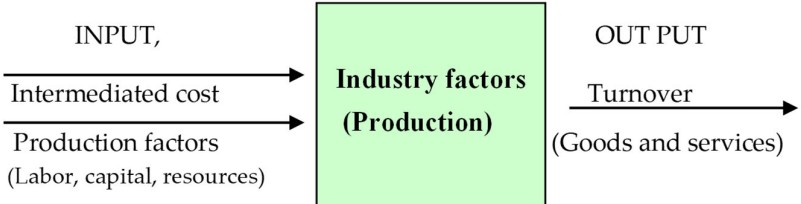

**Chart 1.** Commodity chain analysis flow. Source: Lebailly [55].

This method was exploited to analyze the distribution channel and value added by each actor in the goat chain. This approach also explained how public policies, investments, and institutions affected the goat production systems. It included quantitative analysis of inputs and outputs and the prices and value added throughout the whole goat chain.

Furthermore, the descriptive analysis method was applied to present the basic conditions of the study site, as well as characteristics of the major chain actors through the use of analysis tools presented in graphs and tables in the next section.

## 3. Results and Discussion

*3.1. Goat Production in Lao and Vietnam*

3.1.1. Goat Production in Vietnam

During the period between 2008 and 2018, the total number of goats and sheep produced in Vietnam increased from 1.2 million to 2.8 million heads, with an average annual increase of 8.2%. This demonstrates that goat and sheep production has been very relevant in Vietnam recently. However, the total number of sheep was less than one million in 2018, which was lower than the planned development to reach 3.7 million heads of goats and sheep (Figure 2).

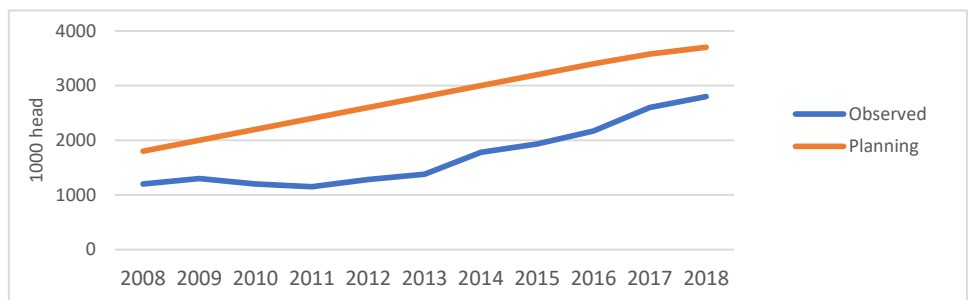

**Figure 2.** Total number of goats and sheep between 2008 and 2018 in Vietnam. Source: Vietnam Department of Livestock Production [56].

Goat production mostly is raised in some provinces in each region (Figure 3), for example in the Northern region: Ha Giang, Nghe An, Son La, and Thanh Hoa provinces; while in the Southern region goats are raised in the following: Ninh Thuan, Dong Nai, Ben Tre, and Gia Lai. Goat production is suitable for the smallholders in the mid- and highland areas, as it can improve the livelihoods for local people [8,9].

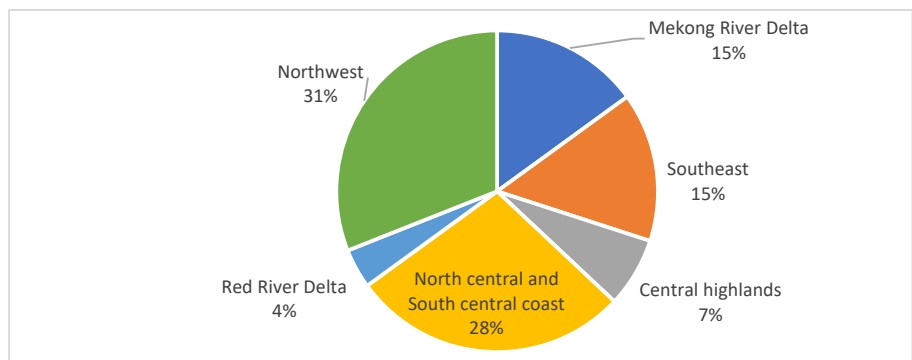

**Figure 3.** Distribution of goats and sheep among regions and subregions of Vietnam in 2018. Source: Vietnam Department of Livestock Production [56].

From 2008 to 2018, goat and sheep meat production increased from 13.5 to 27.5 thousand tons in Vietnam, with an average annual increase of 5.5% (Figure 4). After the period of restructuring animal breeding, some provinces selected goats and sheep as the dominant animals for their areas; hence, the meat production was mostly close to the development orientation number.

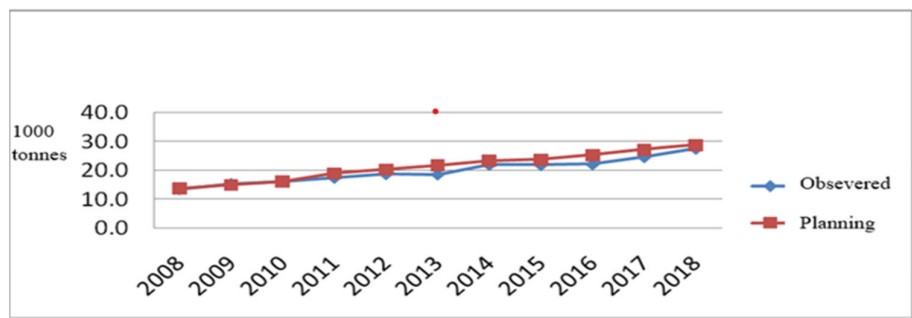

**Figure 4.** Goat and sheep meat production between 2008 and 2018. Source: Vietnam Department of Livestock Production [56].

Given that in 2018 there were only 2.8 million heads of goat and sheep, below the planned amount of at 3.7 million heads, goats were imported from Laos and Cambodia to meet the local market demand [15].

The Ministry of Agriculture and Rural Development achieved an increase in goat and sheep production between 4 and 4.5 million heads, from which 90% of them are goats and crossbred sheep mostly raised at big-scale farms, using a combination of in-stall feeding and controlled grazing systems. However, there have been several challenges in reaching the planned target; hence, there has been an opportunity for goats from Laos to be exported to Vietnam.

### 3.1.2. Goat Production in Laos

In Laos, the free-range system is applied, and goats are grazing all year freely in small groups in the forest and fallow land in mountain areas, as reported by Gray et al. [15]. The Lao Ministry of Agriculture and Forestry established a Livestock Development Plan [57] in which livestock production is expected to be strengthened and promoted. Historically, goat numbers in Lao have been low but have increased rapidly in the last two decades from 121,700 heads in 2000 to 616,325 heads in 2018, an increase of 506.4% (Figure 5). The relatively high rate of increase in the goat population in Laos has been due to a good local market demand for goat meat [15]. In addition, Gray et al. [15] indicated that this growth was driven primarily by an increase in the demand from Vietnam due to both human population growth of 19% in 2000–2016 and, more significantly, a 228% increase in GDP per capita over the same period.

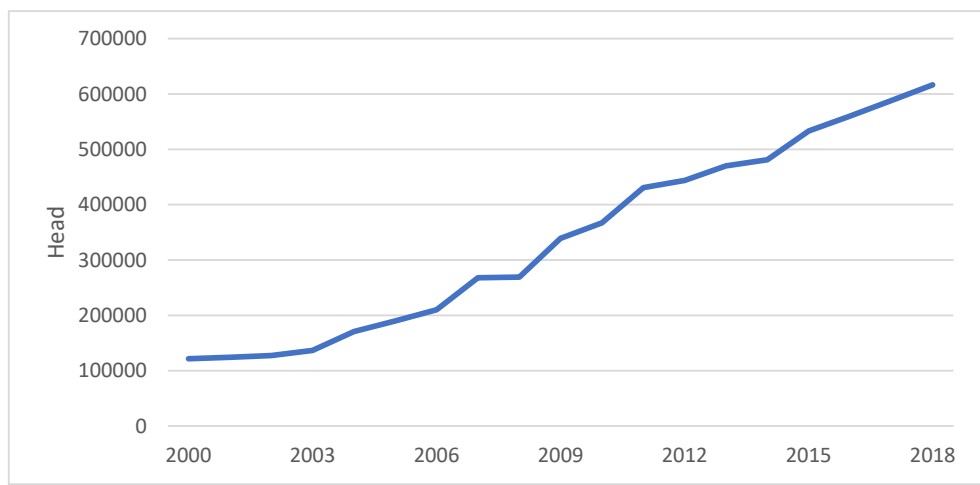

**Figure 5.** Goat production in Laos. Source: FAOSTAT, 2020 [58].

*3.2. Goat Marketing Channels*

Goat production in Laos is mainly for meat. It is consumed locally or sold to local or Vietnamese traders as live animals. Two major goat distribution channels were found. The first one was domestic, where the farmers sold live goats to Lao traders (77.74%) and Lao abattoirs (22.26%). Lao traders, in turn, sold live goats to Lao abattoirs (0.49%) and Lao restaurants (14.35%). Abattoirs and restaurants slaughtered their goats, while most of the goat meat from abattoirs was sold to Lao restaurants, some meat shops, and supermarkets (0.49%) before it was offered to Lao consumers.

The second channel was international distribution, with most of the goats being gathered by Lao traders and sold to Vietnamese traders (85.16%) inside Laos or at the border gates. Then, goats were supplied to Vietnamese abattoirs (59.66%), to restaurants (33.3%), and sometimes directly to the final Vietnamese consumers (7.04%). In some cases, Vietnamese people buy live goats and slaughter them for their events such as weddings, festivals, and New Year, among other festivities. Goat meat from abattoirs was provided mostly to Vietnamese restaurants (99.21%), some meat shops (0.79%), and then to consumers (Figure 6). Most Lao goats served Vietnamese consumers.

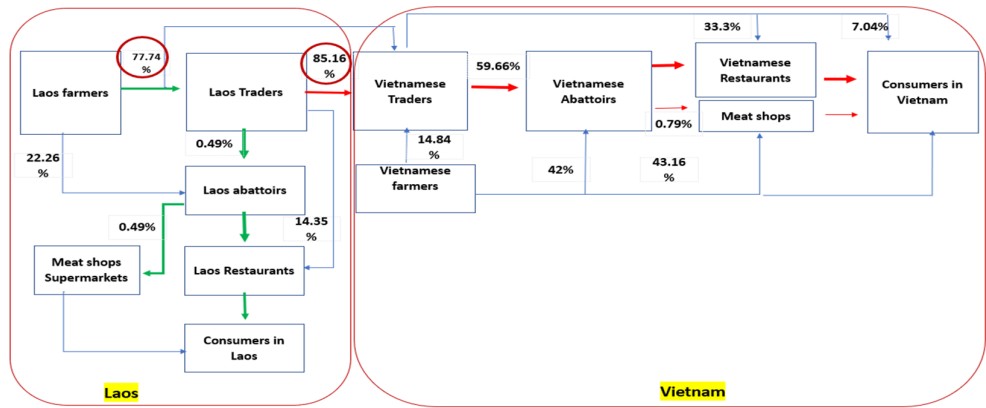

**Figure 6.** Goat distribution channels. Source: Survey data, 2022.

The marketing channel for Lao goats was mostly informal and based on oral agreements, which affected the prices received by farmers. Some previous studies indicated that animal herd size, the gender of the decision maker, and the geographical location are factors that affected the choice of the marketing channel [15,17,18,59,60]. Smallholders with big herd sizes were more likely to sell to traders than to individual smallholders. The choice of the marketing channel used by the farmer had a bearing on the price received

and the gross margins. There was no standardized pricing, and in most of the cases the age, size, and gender of the goat determined the price paid. This scenario contrasts with the outcome of more formal market systems, which have an increase in transparency, leading to a price discovery system that provides incentives and rewards for both buyers and sellers [6]. The goat marketing chain in Laos has constraints to provide the right quantity and quality of goats as demanded by the markets, it lacks uniform carcasses of goats, and it is an unclear market link that can lead to producer exploitation and a lack of or unregulated post-slaughter processes [61].

Formal markets of Lao goats have been poorly developed and, in most cases, are non-existent. Farmers sold goats mainly on a cash-need basis and at giveaway prices, without realizing the true value of the goats. This leads to a lack of incentives to improve productivity and market infrastructure for goats, as described by Dube et al. [49].

Nowadays, there is a combination of production and marketing deficiencies that need to be adequately and properly treated and integrated to solve and improve the performance of the goat value chain. The marketing of live goats and goat meat in Laos was traditional and was mostly handled by private traders. A number of middlemen were involved in the chain of marketing live goats and goat meat. Besides local markets, particularly in rural areas in Laos [62], there has been an increase in goat numbers and demand for goats at restaurants in Laos and in Vietnam, as described by Pathoummalangsy [63].

In the past, Lao goats were purchased by Vietnamese traders on the shared border of Laos and Vietnam, but this has decreased due to Lao government regulations. This has tended to lower the price and make goats more affordable to Lao consumers [63]. Survey results showed that some Vietnamese traders traveled deep into Laos to buy goats for them to be sold in Vietnam. Many goats from remote districts were sold by traders on transport routes in Laos or taken into Vietnam to the larger centers, as shown by ADB [64].

### 3.3. An Analysis of the Goat Value Chain from Laos to Vietnam
3.3.1. Mapping the Goat Value Chain

The goat chain from Laos to Vietnam has four main functions: production, trading, slaughtering, and processing and distribution. These functions are implemented by four prominent actors, starting from goat-rearing farmers; traders (including internal and external traders); abattoir owners; and retailers, including outlets serving goat meat and restaurant owners (Figure 7). In contrast to the case of other ruminant animal chains, such as dairy cattle [65], or the goat value chains in other areas [6,50], the role of input suppliers seems unclear in this chain. There was no clear evidence or report on the goat processing industry in Laos, with the exception of restaurants that processed goat meat to serve to their customers. Similarly, there was no declaration of the leader in creating cohesion and leading actors along the chain.

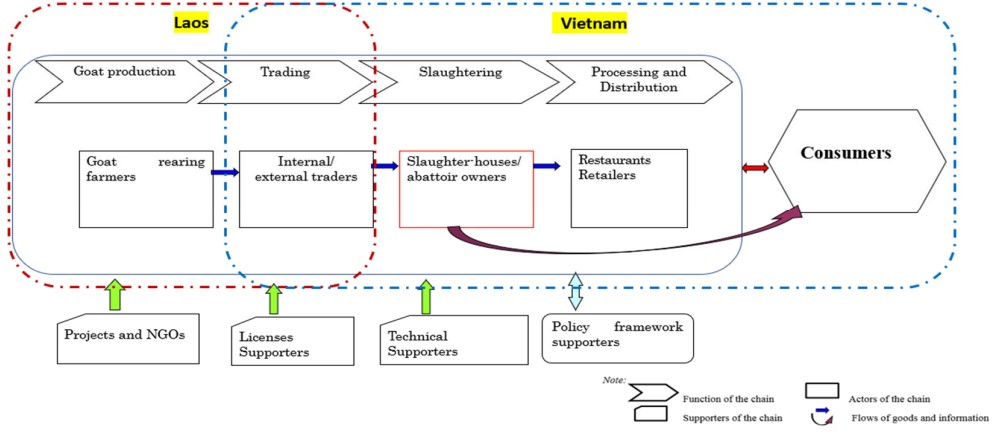

**Figure 7.** The goat value chain from Laos to Vietnam. Source: Survey data, 2022.

### 3.3.2. Analysis of Goat Chain Actors

Farmers were the first actor of the goat chain; however, their role and price negotiation power was low. Most of them were independent farmers and did not work under the control of other actors along the chain. Therefore, farmers reared goats without support from traders, abattoir owners, and restaurant owners. All surveyed farmers were smallholders, which is similar to the result previously reported by Stur and Phengsavanh [59], with the median and average number of goats per householder of nine and eleven heads, respectively. The largest scale farm had 45 heads, showing a slow-growing trend in the rearing scale compared to previous research, ranging from 2.2 to 3.4 heads in 2003 [66] to 5.3 heads in 2013 [67] and 10.2 in 2018 [15]. Most of the goats originated from Lao goats, and most of the goat breeds came from a natural re-herd (Table 3).

**Table 3.** Goat production.

|  | Head/Farm | On-Farm Origin (%) | Value (1000 Kips [*]/Head) |
|---|---|---|---|
| Kid (<6 months) | 4.05 | 99.47 | 347.46 |
| Yearlings (6–<12 months) | 2.65 | 99.19 | 701.37 |
| Bucks (>12 months) | 0.69 | 92.41 | 1160 |
| Ewes (>12 months) | 3.61 | 94.88 | 1057.8 |

Source: Survey data, 2022 (*) Kip is Lao's currency.

The primary purpose of keeping goats was selling due to high demand from the market (35%) and high profit (22%) (Figure 8). This was also found by Bounthavone et al. [68]. In previous studies, results from Houaphanh, Luang Prabang, and other northern provinces in Laos showed that 90 to 95% of goats were kept for exporting to Vietnam [15]. The demand for goat meat was constant throughout the year and goats can be sold at any time, even if animals are in poor condition, which is common across many goat-producing countries [69]. There are well-organized systems for selling cattle and buffalo [60], but processing is unclear for Lao goats.

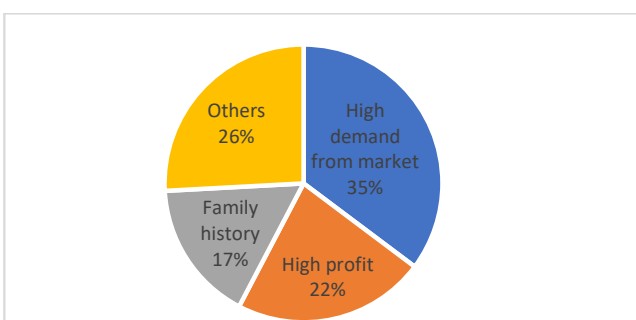

**Figure 8.** Farmers' motivation for goat rearing. Source: Survey data, 2022.

Goats were sold mainly to Lao traders (63%) and aggregators (26%) (Figure 9). Goats were an excellent alternative for farmers in poor areas to access high-value markets, because market demand was strong, capital investment was low, diseases were few, they were adaptable to a wide range of feeds, and the reproductive rate of kids was high.

Aggregators collected and sold goats without fattening. However, traders collected goats and then sold them if they obtained a suitable price; otherwise, they kept and fattened their goats before selling them domestically or internationally. They sold live goats to other Lao traders, restaurants, or outlets serving goat meat for domestic trading, and they sold unofficially to Vietnamese traders near Laos and Vietnam's shared border for international trading. Vietnamese traders purchased goats and sold or fattened goats before selling them to Vietnamese abattoirs and restaurants, and some were sold directly to consumers. This is

why the average live weight of goats purchased from traders (30.1 kg) was heavier than those from farmers (22.5 kg) or aggregators (23.8 kg) (Table 4). Most goat traders worked independently, and only some of them depended on the slaughterhouses. This outcome is similar to the results reported by Gray et al. [15].

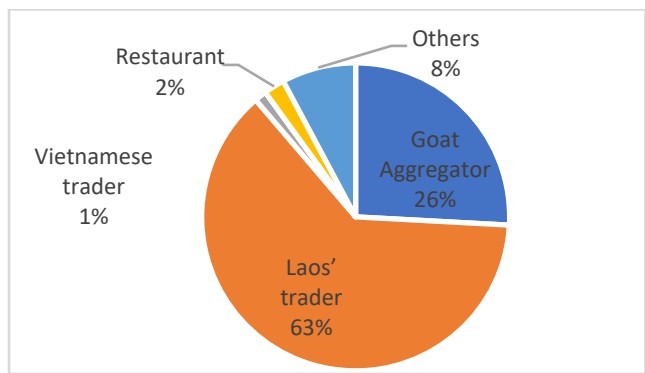

**Figure 9.** Goat selling. Source: Survey data, 2022.

**Table 4.** Quantity of goats purchased by a trader.

| Sources | Quantity per Week | | Average Live Weight |
|---|---|---|---|
| | **Head** | **%** | **(kg)** |
| From farmers | 49 | 23.2 | 22.5 |
| From other traders | 148 | 69.6 | 30.1 |
| From aggregators | 15 | 7.19 | 23.8 |

Source: Survey data, 2022.

The abattoir owners purchased goats from farmers (42.4%) and traders (57.6%) due to the clear origin and ease of buying, with an average live weight of 24 kg per goat (Table 5). Most abattoir owners were small-scale, with a capacity of slaughtering 3.4 heads per day on average. The number of slaughtered goats depended on seasons and market demand, with some abattoirs slaughtering only from 10 to 15 heads per month. After slaughtering the goats, they sold most of the carcasses or meat cuts to restaurants, as is reported in previous research studies [15]. A small portion of goat meat was sold in local markets, where consumers could buy and prepare it at home.

**Table 5.** Goats for slaughtering at the abattoir.

| Sources | Quantity | | Average Live Weight | Reasons to Purchase Goat (%) | | | |
|---|---|---|---|---|---|---|---|
| | **Head/Day** | **%** | **(kg)** | **Origin** | **Easy** | **Low Price** | **Late Payment** |
| From farmers | 10.1 | 42.4 | 24.1 | 38.89 | 30.56 | 8.33 | 0 |
| From other traders | 13.7 | 57.6 | 24.2 | 44.44 | 55.56 | 0 | 2.78 |

Source: Survey data, 2022.

In Laos, there were only some restaurants in each large town, and from 30 to 40 outlets serving goat meat in Vientiane to domestic consumers [15]. Meanwhile, in Vietnam, there were many restaurants and retailers selling meals with goat meat. Hence, Vietnam is considered a large consumption market for Lao goat meat. The Lao goat, referred to in Vietnam as a 'mountain' goat, 'local' goat, or 'grass' goat, receives a significant premium in the Vietnamese restaurant trade. This perception was also reported by Gray et al. [15].

Restaurants bought live goats and slaughtered them or purchased goat meat to process and serve their customers. In the case of live goats, they were primarily purchased from traders (15 heads/week) and some from farmers (6 heads/week). Some bought carcasses (15.2 kg/week) or meat cut (49.6 kg/week) from abattoirs (Table 6). There were around twenty-five dishes of goat meat sold in the local restaurants, from them about ten were popular among consumers, with goat restaurants usually crowded. The survey also showed that Vietnamese people liked goat meat and mainly ate it at restaurants.

**Table 6.** Quantity of weekly goat processed in the surveyed in restaurants.

| | Live Goat | | Carcass | Meat Cut |
|---|---|---|---|---|
| | **Quantity** | **Average Live Weight** | | |
| | **Head** | **kg** | **kg** | **Kg** |
| From farmers | 6.2 | 24.9 | | |
| From traders | 14.9 | 30.8 | | |
| From abattoirs | | | 15.2 | 49.6 |

Source: Survey data, 2022.

There were both vertical and horizontal linkages in the Lao goat chain. There were 61.11% of traders who had linkages with other actors, including other traders (69.57%), farmers (43.48%), and abattoirs (8.70%), mainly using oral contracts. A total of 44.44% of abattoirs had a linkage with farmers and traders, of which nearly half had written contracts. A total of 65.42% of restaurant owners had a linkage with other actors, mainly through oral contracts. Due to the informalities along the chain, there was no evidence of a traceability system that had been implemented.

### 3.3.3. Supporting Stakeholders of the Goat Chain

The four main supporting stakeholders of the goat chain from Laos to Vietnam were government and local authorities; projects and NGOs; technicians; and license supporters. Lao goat husbandry has been regulated by The Lao Ministry of Agriculture and Forestry, the Department of Livestock and Fisheries at the national level, the Provincial Agriculture and Fisheries Office, and the District Agriculture Forestry and Extension Office [8,16].

The Asian Development Bank [64], the Australian Centre for International Agricultural Research [15], the National Agriculture and Forestry Research Institute [17], the International Center for Tropical Agriculture, the International Livestock Research Institute, the Food and Agriculture Organization of United Nations, and other international and national agricultural research institutions from countries including Australia and Sweden [15] support promoting and strengthening goat production through improving the capacity for the local people. However, different from other goat value chains in other countries [50], the credit service did not appear in the chain.

Veterinarians and livestock scientists support goat management, breeding, disease control, and goat protection from harsh conditions or protection from suffering the effects of natural disasters [69,70].

License supporters provide three kinds of certificates [18], including a Health Certificate—obtained within the district of origin and checked by the District Agriculture and Forestry and Extension Office officers, a Certificate of origin that is purchased at the point of origin, and an export license, which is purchased by accredited companies from the government.

### 3.4. A Socio-Economic Sustainable Development Perspective

3.4.1. Economic Sustainability

By the time of the survey, after around one year of rearing, goat-rearing farmers in Laos earned on average 2.87 USD (and a maximum of 3.11 USD) per kg of the live goat; from that amount, half was the cost of production, including breed, feed, vaccination, veterinary

medicine, etc. The economic benefits of goat-rearing farmers in Lao are much higher than those of the goat chain, as reported by Hussen et al. [50] in Ethiopia, Thelma et al. [6] in Zambia, and Sarker et al. [47] in Bangladesh. They could even obtain higher incomes from goat rearing by adopting the scientific norms of goat management [71] and potentially higher by improving the herd size and engaging in goat rearing as an enterprise.

Lao traders sold 85% of the goats to Vietnamese traders at 5.28 USD (and in some cases, they reached 5.35 USD) per kg of live weight, much higher than those sold to Lao abattoirs and restaurants at 3.3 USD. On average, Lao traders obtained 8.26% of profits from goat trading; however, traders in Laos obtain a lower economic benefit in comparison to Vietnamese traders, which is similar to the cases reported by Hussen et al. [50] in Ethiopia, and Thelma et al. [6] in Zambia.

Lao restaurants mainly purchased live goats from Lao traders, and only some purchased goat meat from abattoirs at an equivalent price of 4.85 USD per kg of carcass weight. On average, they earned a profit of 30.92 USD per live goat, equivalent to 1.55 USD per kg of a live goat.

In the case of Vietnamese traders, they bought live goats from Lao traders and sold them mainly to Vietnamese abattoirs and restaurants at 5.95 USD per kg, achieving a profit of 7%. Furthermore, Vietnamese abattoirs sold goat meat to restaurants at the price of 7.53 USD per kg of the carcass, making a profit of 7.73%.

Finally, some Vietnamese restaurants purchased live goats from traders, while others bought goat meat from abattoirs to sell goat meat to their customers, mainly restaurant consumers, and achieved profits of 1.58 USD per kg of live goat equivalents (21%) (Figure 10). The economic benefits of the goat chain actors from Lao to Vietnam are much higher than those of the goat chain reported by Hussen et al. [50] in Ethiopia and Thelma et al. [6] in Zambia.

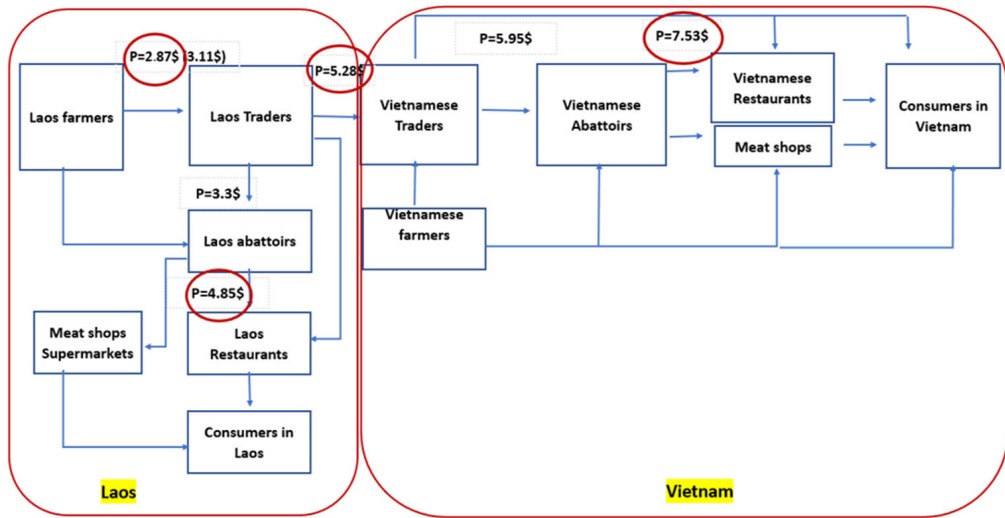

**Figure 10.** Prices and value added throughout the goat chain from Laos to Vietnam. Source: Survey data, 2022.

The results showed that all actors benefited from the goat chain, especially from Laos to Vietnam. It also showed that goat-rearing farmers obtained a high added value proportion. However, the profit was lower if we calculated the family's labor cost, farmland opportunity cost, and goat investment in breed and barns. In the Lao goat value chain, traders and restaurant owners seemed to benefit more from the chain.

### 3.4.2. Social Sustainability

Besides the economic aspects, the goat chain from Laos to Vietnam also showed the opportunity for a socially sustainable development, because this chain has created job opportunities and income to improve living standards for all chain actors, especially for

disadvantaged groups in society, such as women, the middle-aged, people with low levels of literacy, and people in rural areas.

The characteristics of each stage in the chain were different in their characteristics and distribution of jobs and incomes across the actors. In the goat production and trading stages, most of the work was implemented in rural areas, mainly by women. This result was quite similar to those reported by Islam et al. [45] and Chowdhyry et al. [44] but different from other previous studies conducted by Hussen et al. [50] and Thelma et al. [6]. The role of male and female labor was not different from the case in other areas [50]. The educational level was mainly under high school, similar to previous research [6]. The actors in the stages at the end of the chain, the abattoirs and restaurant owners, were mostly carried out in urban areas rather than in rural areas, with the male labor force predominating. However, females still accounted for a proportion of around 35%–40%. The education level of these actors was higher than previous groups, but they were still mainly unskilled, as nearly 70% of them had high school and under high school educational levels.

In the whole chain, there were job opportunities created and income for 62% of actors in the rural areas, compared to 38% of those in the urban area. Job creation and income for female actors was 56.25%, higher than that of 43.75% for males. Most of them were in their middle age, with an average of 44.23 years old. The chain not only created jobs and income for the chain actors but also generated an average of 3.41 working laborers per actor (except goat-rearing farmers), and of those, there were 1.9 female laborers (55.7%), with an average income of 163.92 USD per month for their permanent work, in addition to creating income and jobs for the temporary workforce (Table 7).

**Table 7.** Socio-economic characteristics of chain actors.

| Criteria | Farmers | | Traders | | Abattoir | | Restaurant Owners | |
|---|---|---|---|---|---|---|---|---|
| | Freq. People | Percent (%) | Freq. People | Percent (%) | Freq. People | Percent (%) | Freq. People | Percent (%) |
| Total | 221 | 100 | 36 | 100 | 36 | 100 | 107 | 100 |
| Rural | 187 | 84.62 | 29 | 80.56 | 15 | 41.67 | 17 | 15.89 |
| Urban | 34 | 15.38 | 7 | 19.44 | 21 | 58.33 | 90 | 84.11 |
| Gender | | | | | | | | |
| Male | 78 | 35.29 | 5 | 13.89 | 22 | 61.11 | 70 | 65.42 |
| Female | 143 | 64.71 | 31 | 86.11 | 14 | 38.89 | 37 | 34.58 |
| Education | | | | | | | | |
| Under high school | 127 | 57.99 | 21 | 58.33 | 12 | 33.33 | 15 | 14.02 |
| High school | 28 | 12.79 | 13 | 36.11 | 17 | 47.22 | 56 | 52.34 |
| College | 12 | 5.48 | 1 | 2.78 | 4 | 11.11 | 17 | 15.89 |
| University | 2 | 0.91 | 1 | 2.78 | 2 | 5.56 | 18 | 16.82 |
| Other | 50 | 22.83 | | | 1 | 2.78 | 1 | 0.93 |
| | Obs | Mean | Obs | Mean | Obs | Mean | Obs | Mean |
| Age | 221 | 46.15 [a] | 36 | 44.67 [a] | 36 | 39.39 [a] | 107 | 41.76 [a] |
| Working labor | 221 | 1.40 [b] | 36 | 2.78 [b] | 36 | 3.14 [b] | 107 | 7.89 [b] |
| Working female labor | 221 | 0.81 [b] | 36 | 0.67 [b] | 36 | 1.61 [b] | 107 | 4.64 [b] |
| Permanent labor cost | | | 16 | 227.94 [c] | 25 | 151.30 [c] | 99 | 221.34 [c] |
| Temporary labor cost | | | 13 | 372.62 [d] | 12 | 1776.20 [d] | 92 | 703.22 [d] |

Source: Survey data, 2022. Note: (a): Year old; (b): People/farm; (c): USD/month; (d): USD/year.

## 4. Conclusions

Goats thrived in Laos under extensive conditions with minimal management and have a high potential for further development. The Lao Ministry of Agriculture and Forestry established the livestock development plan aiming to strengthen and promote livestock production. Historically, the number of goats in Lao has been low, but it has increased rapidly in the last two decades, primarily driven by the demand from Vietnam.

This study reviewed the literature and found several studies on value chains in general and on goat value chains in particular. Value chain analysis is valuable for new producers, especially for poor producers in poor countries to achieve sustainable income growth, but there has been no previous research on the goat value chain from Laos to Vietnam. This study analyzed this case study and found that the marketing channel for goats was mostly informal and underdeveloped in Laos. While local markets remained important, particularly in rural areas in Laos, an increase in the goat meat demanded by restaurants in Laos and Vietnam has contributed to the development of this market chain. There were two basic marketing channels for goats: the domestic marketing channel and the export marketing channel, mainly to Vietnam.

The Lao goat chain had four main functions: production, trading, slaughtering, and processing and distribution. These functions were implemented by four prominent actors along the chain, starting from goat-rearing farmers, traders (including internal and external traders), abattoir owners, and restaurant owners. The role of input suppliers seemed to be unclear in this chain. Production was mainly based on a small scale. Traders worked as goat collectors and at fattening, while there was no evidence or report on the goat processing industry. The four main groups supporting stakeholders for the goat chain in Laos were policy framework supporters, projects and NGOs, technical supporters, and license supporters.

There were both vertical and horizontal linkages in the Lao and Vietnam goat chain, but mainly oral agreements prevailed. There was no declaration of a leader in creating cohesion among actors, and there was no indication of leading actors along the chain. In addition, there was no clear evidence of a traceability system, even though it could be potentially appreciated by consumers in Laos and Vietnam. This type of export market demand-driven chain has developed based on the high goat meat demand from Vietnamese consumers.

The most important aspect of the analysis found both in practice and theory is the chain-created sustainable economic and social benefits for the chain actors along the chain. The commercialization of goats yields positive net income in Laos, and all chain actors benefited from their participation in the chain, from which traders and restaurant owners gained the most. Furthermore, the chain also created job opportunities and income to improve living standards for all chain actors, especially for disadvantaged groups such as women, the middle-aged, people with low levels of literacy, and those living in rural areas. This is a significant condition for the chain to develop sustainably because when the actors gain the benefits, they have the motivation to continue their job and the chain will exist for a long time.

In order to develop the goat chain from Laos to Vietnam with a sustainable approach, some recommendations are provided as follows:

- The Lao and Vietnamese governments should create more favorable conditions for the development of the goat chain by improving the legal framework status and licensing support for the official export of goats from Laos to Vietnam.
- The Lao government could develop policy programs to promote goat husbandry, plan areas for growing goat feeding, and train farmers to raise awareness on feeding supplementation to ensure adequate quality feeding for goats.
- Goat-rearing farmers should upgrade their herd management capacity to improve efficiency and sustainability in livestock production. They can learn and share goat-farming experiences among smallholders in the region to enhance their knowledge and skills in raising goats more efficiently.

- Other chain actors could have connections with each other to develop formal linkages and official marketing channels and to improve traceability across the goat chain. This could increase the benefits for each actor and the sustainability of the chain.
- Finally, in the case of other supporting stakeholders, training institutions, agricultural extension organizations, animal veterinarians, and non-governmental organizations should provide technical support, solutions based on research studies, and technologies that encourage farmers and local authorities to implement applied sciences in goat-rearing and capacity building.

**Author Contributions:** T.N.B., H.V.N., X.B.N., V.N.L., T.M.N., C.T.K.N., Q.T.L.N., N.H., L.E.M., V.D.N., L.O. and S.W.-B.: conceptualization, methodology, supervision; T.N.B., H.V.N., X.B.N., V.N.L., T.M.N., C.T.K.N., Q.T.L.N. and V.D.N.: investigation; T.N.B.: writing—original draft preparation, validation, formal analysis, resources and data curation, writing—review, and editing; N.H., L.E.M., L.O. and S.W.-B.: project administration; N.H. and L.E.M.: writing—review and editing; T.T.H.L.: software, methodology, data curation, validation. All authors have read and agreed to the published version of the manuscript.

**Funding:** This research was funded by the Australian Centre of International Agricultural Research (project number: LS/2017/034).

**Institutional Review Board Statement:** The study was conducted in accordance with the Declaration of Helsinki, and approved by the Human Research Ethics Committee of University of New England (protocol code HE20-208 and date of approval of 12/08/2021).

**Informed Consent Statement:** Not applicable.

**Data Availability Statement:** In case of need, please contact corresponding author for the data.

**Conflicts of Interest:** The authors declare no conflict of interest.

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
