# Peer review of "An Analysis of the Goat Value Chain from Lao PDR to Vietnam and a Socio-Economic Sustainable Development Perspective"

_sustainability, doi:10.3390/su151813781_

Round 1

Reviewer 1 Report

SPECIFIC COMMENTS:

Abstract

On Line 36 says: sustainably. I suggest using sustainable

Keywords: The keywords Sustainable development perspective; goat value chain; Lao PDR and Vietnam are in the article title. Therefore, I suggest changing them to some of the following: goat rearing; goat meat; small scale production; traders; marketing; abattoirs; restaurants.

Introduction

No comments

Materials and methods

Page 3. The source clarifying note in Figure 1 appears as page footnote. I suggest that it be placed as Figure footnote.

On Page 3, Line 119 says: For farmers, the sample size was selected … ; should say: For farmers, the sample size (SS) was selected …

On Lines 143-144 says: 400 samples were collected, including 265 samples from Laos and 135 from Vietnam, of which there were 221 farmers, 36 traders, 36 abattoirs, and 107 restaurant owners. It should say: Information was collected from 400 people, 265 from Laos and 135 from Vietnam, of which there were 221 farmers, 36 traders, 36 slaughterhouse owners and 107 restaurant owners (Table 1).

Table 1 is not cited in the text. I suggest that the authors cite this Table in the corrected paragraph on lines 143-144.

Lines 151-152. I suggest that the authors include in this lines the Lebailly citation on which the commodity chain analysis methodology is based.

On Line 152 says: was calculated by an equation: …; I suggest changing to: was calculated by the equation: …

Results and discussion

Tables 1 to 6 are not cited in the text. I suggest that the authors cite each Table in the appropriate place of the corresponding paragraphs.

The Figure 1 title is duplicated. This title is entered on line 116 (Map) and line 184 (Goat production in Laos). On the other hand, in line 179 Figure 1 is cited, so it is not clear what is the correct title of Figure 1. Based on the above, I suggest that the authors verify the Figure 1 citation in the text, correct and renumber the title of the Figure Goat Production in Laos (Line 184) and all subsequent Figures titles.

Figures 2 to 5 are not cited in the text. I suggest that the authors cite each Figure in the appropriate place of the corresponding paragraphs.

On Page 5, superscript 2 of the page footnote is not cited in text. The authors must cite this superscript in the text.

On Page 6, Line 264, the superscript 3 noted in Table 2 is not cited as table footnote, it is cited as page footnote. I recommend that the authors cite this superscript in Table 2 as table footnote.

On Line 270 says: Goats an be sold … ; should say: Goats can be sold …

On Line 314 says: werepopular ...; should say were popular...

Page 10, Line 367. The Figure title is misaligned in the text. Please correct.

On Line 394 says: highs chool … ; should say: high school...

In Table 6 (Page 11), the explanation of the superscripts 4, 5, 6 and 7 does not appear in the table footnote. The explanation of these superscripts appears as a footnote on pages 11 and 12, respectively, so the explanation of the superscripts in the Table 6 footnote must be corrected.

On Line 431, I recommend using sustainable instead of sustainably

On Line 346 it was written: 3.4.1. A sustainable in the economic aspect, I suggest changing to 3.4.1. Sustainability in the economic aspect or simply 3.4.1. Economic sustainability.

On Line 375 it was written: 3.4.2. Sustainable in the social aspects. I suggest changing to 3.4.2. Sustainability in the social aspects or simply 3.4.2. Social sustainability.

Sections 3.4.1. (Page 10) and 3.4.2. (Page 11). The paragraphs of both sections do not include bibliographic citations; therefore, I suggest that the authors include bibliographic citations for each of the topics covered in both sections (Economic sustainability and Social sustainability, respectively).

Conclusions

On Page 12, Lines 426-430. I suggest that the authors include in the writing of the paragraph the concepts of economic and social sustainability.

On Line 434 says: the goat chainby...; should say: the goat chain by …

References

The writing of all bibliographical references must be uniform. Specifically, authors should review and correct the writing of references 15 and 18.

Author Response

Please see our response in the attached file. Thank you.

Reviewer 2 Report

This study aimed to analyze the goat value chain from Laos to Vietnam and a socio-economic sustainable development perspective, based on 400 samples of data collected using standard questionnaires with the support of CommCare software from chain actors in the second and third quarters of 2022. The authors have investigated an interesting research question. I’m concerned with the following issues in the study.

Major concerns:

1. Provide a table to analyze the literature review. What are the research gaps? How can you fill them? Compare the existing models in the literature and why you think your model is merit in the literature? Please work on improving the clarity of your paper.

2. On Page 12, in the part of conclusion, the author spent a lot of time sorting out the research findings. I do not find the theoretical contributions of the research to the existing research field.

3. If you want to pick out the most one finding in this paper, please show this result, explain the intuitions, and clarify its important in both practice and theory.

Minor concerns:

1. On Page 2, at the end of introduction, the organization structure of this paper should be added.

2. Some recent papers about value chain management should be added into the literature review. The following related papers are suggested: Uncertainty analysis and optimization modelling with application to supply chain management: A systematic review.

3. Writing: There are still some typos, mistakes, and grammar issues throughout the entire paper.

Minor editing of English language required.

Author Response

(The authors gave the same response as above.)

Reviewer 3 Report

Dear Authors and colleagues,

Glad to hear good news from you. I was appointed by the Editor to review the paper you submitted regarding an investigation of the value chain in goat commodities with case studies in two countries (Laos and Vietnam). In the economic context, especially business health, my first observation was very enthusiastic about the contents of the paper, which highlighted the role of four actors (goat breeders, traders, slaughterhouse owners, and restaurant owners) that are predicted to have a systematic effect on the sustainability of local livestock. Theoretically, the orientation for the success of the goat farming business is a broad export market and produce a clear system, but based on the identification of this study, the results are contradictory. With a low intensity of commercialization, it actually has an impact on employment and income, which are the most crucial and have the risk of affecting the standard of living of the farmer group. Through my current observations, I see that there are technical problems in writing papers. Therefore, I highlighted and commented on 15 points for revision. Each corrected part must be applied specifically. Good luck.

All the best,

Anonymous reviewer

Author Response

(The authors gave the same response as above.)

Round 2

Reviewer 1 Report

Thank you for attending the suggested corrections.